Alpine endemic spiders shed light on the origin and evolution of subterranean species

Mammola Stefano 1
Isaia Marco 1 marco.isaia@unito.it
Arnedo Miquel A. 2
1 Department of Life Sciences and Systems Biology, University of Turin , Turin , Italy
2 Departament de Biologia Animal & Biodiversity Research Institute, Universitat de Barcelona , Barcelona , Spain
Stanford Jack
Electronic publication date: 2015 Nov 3
Publication date: 2015
Volume: 3
Electronic Location ID: e1384
Received 2015 Aug 22; Accepted 2015 Oct 15
Copyright: © 2015 Mammola et al.
Copyright year: 2015
Copyright holder: Mammola et al.
License: This is an open access article distributed under the terms of the Creative Commons Attribution License, which permits unrestricted use, distribution, reproduction and adaptation in any medium and for any purpose provided that it is properly attributed. For attribution, the original author(s), title, publication source (PeerJ) and either DOI or URL of the article must be cited.
License URL: https://creativecommons.org/licenses/by/4.0/

Keywords: Comparative phylogeography, Pleistocene glaciations, Ecological niche modeling, Cave-dwelling spiders, Alpine fauna, Pimoa, Troglohyphantes, Subterranean specialization, DNA markers, Dispersal

Funding: University of Turin and Compagnia di San Paolo (Progetti di Ricerca di Ateneo 2011—ORTO11T92F) This study was set within the CaveLab project “From microclimate to climate change: caves as laboratories for the study of the effects of temperature on ecosystems and biodiversity”, funded by University of Turin and Compagnia di San Paolo (Progetti di Ricerca di Ateneo 2011—ORTO11T92F). The funders had no role in study design, data collection and analysis, decision to publish, or preparation of the manuscript.

==============================
We designed a comparative study to unravel the phylogeography of two Alpine endemic spiders characterized by a different degree of adaptation to subterranean life: Troglohyphantes vignai (Araneae, Linyphiidae) and Pimoa rupicola (Araneae, Pimoidae), the latter showing minor adaptation to hypogean life. We sampled populations of the model species in caves and other subterranean habitats across their known geographical range in the Western Alps. By combining phylogeographic inferences and Ecological Niche Modeling techniques, we inferred the biogeographic scenario that led to the present day population structure of the two species. According to our divergent time estimates and relative uncertainties, the isolation of T. vignai and P. rupicola from their northern sister groups was tracked back to Middle–Late Miocene. Furthermore, the fingerprint left by Pleistocene glaciations on the population structure revealed by the genetic data, led to the hypothesis that a progressive adaptation to subterranean habitats occurred in T. vignai, followed by strong population isolation. On the other hand, P. rupicola underwent a remarkable genetic bottleneck during the Pleistocene glaciations, that shaped its present population structure. It seems likely that such shallow population structure is both the result of the minor degree of specialization to hypogean life and the higher dispersal ability characterizing this species. The simultaneous study of overlapping spider species showing different levels of adaptation to hypogean life, disclosed a new way to clarify patterns of biological diversification and to understand the effects of past climatic shift on the subterranean biodiversity.

Introduction

Long term climatic changes are often invoked among the most important factors that drove surface-dwelling invertebrate populations to colonize subterranean habitats, causing their isolation and shaping their present day distribution patterns (Jeannel, 1943; Peck, 1980; Holsinger, 1988; Botosaneanu & Holsinger, 1991; Culver & Pipan, 2010). In this regard, the Pleistocene glaciations (reviewed in Culver & Pipan, 2010) and the Messinian Salinity Crisis (Faille et al., 2010) have been pointed out among the main drivers of the present distribution patterns of the European subterranean biodiversity. The climate-driven isolation caused the allopatric divergence of subterranean populations, resulting in narrow patterns of distribution and high levels of endemism (Christman et al., 2005; Culver & Pipan, 2009; Borges et al., 2012; Cardoso, 2012). Accordingly, population studies conducted so far have uncovered an extreme genetic structuring in terrestrial invertebrates in subterranean habitats (Kane, Barr & Badaracca, 1992; Allegrucci, Minasi & Sbordoni, 1997; Hedin, 1997; Gentile & Sbordoni, 1998; Hedin & Thomas, 2010; Ribera et al., 2010; Snowman, Zigler & Hedin, 2010; Dixon & Zigler, 2011; Zhang & Li, 2013; see also Bohonak, 1999 for an historical perspective on this topic). This general trend was mainly interpreted in light of the past climatic transition, as the result of the limited dispersal ability of subterranean organisms. Because of the adaptation to the hypogean medium, subterranean species develop narrower physiological tolerance (i.e., troglomorphism sensu Christiansen, 1962), which hamper their dispersal ability through non-subterranean habitats. For example, some cave-dwelling spiders with minor adaptations to hypogean life are known to disperse outside caves in different stages of their life cycle (e.g., Meta spiders; Smithers, 2005; Mammola & Isaia, 2014). On the other hand, subterranean habitats, especially caves, are generally connected through a networks of small cracks and voids which may facilitate dispersal of the invertebrate fauna (Juberthie, Delay & Bouillon, 1980; Juberthie, Delay & Bouillon, 1981; Uéno, 1987; Romero, 2012; Culver & Pipan, 2009; Culver & Pipan, 2014; Giachino & Vailati, 2010 among others).

Here we designed a comparative study aimed at providing insights on the origin and the evolution of the hypogean biodiversity. Specifically, we focused on two Alpine endemic species co-occurring in caves and other subterranean habitats across most of their known distribution range and exhibiting different levels of adaptation to subterranean life.

The first model species,Troglohyphantes vignai Brignoli, 1971 (Araneae, Linyphiidae), is endemic to the Western Italian Alps (NW Italy), being discontinuously distributed from the Cottian (Province of Torino) to the Maritime Alps (Province of Cuneo) (Isaia et al., 2011). Since the description, all available records of this species refer to cave habitats (Brignoli, 1971; Brignoli, 1985; Pesarini, 2001; Arnó & Lana, 2005; Isaia & Pantini, 2010; Isaia et al., 2011). T. vignai shows adaptations to the hypogean life, namely loss of pigmentation, reduction of the eye apparatus, thinning of integuments and heavy spination. T. vignai was described from the cave Buco di Valenza (Speleological cadastrial number: 1009 Pi/CN; Po Valley) by Brignoli (1971). In the same publication, Brignoli also described T. rupicapra, which was distinguished from T. vignai by small morphological details of the epigynum. T. rupicapra was described on material from Grotta superiore delle Camoscere (Speleological cadastrial number: 250 Pi/CN; Pesio Valley). According to the species description, and as later observed by Isaia & Pantini (2010, Figs. 15–18), T. rupicapra shows a higher degree of troglomorphism compared to T. vignai, namely higher depigmentation, reduction of eye diameter and loss of functional eyes, and lowering of the profile of the cephalothorax. The species validity of T. rupicapra was questioned by Pesarini (2001), who proposed the synonymy T. rupicapra = T. vignai, currently accepted in the World Spider Catalog (2015).

Our second model organism, Pimoa rupicola (Simon, 1884) (Araneae, Pimoidae), is an Alpine-Apenninic endemic element, recorded almost continuously from the Graian Alps to the Tuscan Apennines (Thaler, 1976; Hormiga, 1994; Isaia et al., 2011) and French Maritime Alps. Several authors (Brignoli, 1971; Brignoli, 1972; Brignoli, 1985; Thaler, 1976; Arnó & Lana, 2005; Isaia et al., 2011) referred to P. rupicola as a troglophile species (sensu Sket, 2008), being abundant in subterranean habitats and occasionally recorded from surface habitats, such as leaf litter, humid rocks covered by mosses and mountain screes (Bertkau, 1890; Jackson, 1926; Thaler, 1976; Hormiga, 1994; Isaia et al., 2015; Isaia, Paschetta & Chiarle, 2015). Given the sporadic collection of individuals outside cave (mainly pitfall trap data reported in Isaia et al. (2015); Isaia, Paschetta & Chiarle (2015), and additional unpublished data collected by two of us (SM and MI)), it seems likely that males and immatures of P. rupicola disperse trough the epigean environment. Morphologically, the species does not show any remarkable troglomorphic features: it has eight functional eyes, it is entirely pigmented and it has a well defined abdominal pattern.

To present knowledge (Arnó & Lana, 2005; Isaia et al., 2011), the Alpine range of P. rupicola encompasses the entire range of T. vignai and the two species often co-occur in the same cave.

In this contribution we investigated the biogeographic events that shaped present day population structure of the two species. Since the study was set at the species/population interface, we employed two fast-evolving DNA markers, namely the mitochondrial cytochrome oxidase I (cox1) and the nuclear second internal transcribed spacer region (ITS-2). The popularity of this markers stem from a generally high level of variation, which permit to reconstruct relationships among and within spider species, making them particularly suitable for population and biogeographic studies (Agnarsson, 2010; Vidergar, Toplak & Kuntner, 2014).

Moreover, in accordance with Peterson (2009) we coupled genetic inferences with ecological niche modeling techniques, thus obtaining multiple supports to our research hypothesis. In particular, we hypothesized that past climatic changes played a key role in shaping the genetic structure of the populations of the two species. Given the contrasting degree of subterranean specialization exhibited by the two spiders, we further hypothesized that populations of P. rupicola show minor genetic structure than T. vignai.

Additionally, this study offered the opportunity to reveal the existence of cryptic species within the two lineages. In fact, it was observed during the course of the study that individuals belonging to the northern populations of P. rupicola presented subtle but consistent differences in their genital morphology. Hereinafter, we will restrict the use of the epithet ‘rupicola’ to indicate the southern populations, and we will refer tentatively to the northern populations as ‘n. sp.’

Material & Methods

Sampling

Populations were collected in caves, abandoned mines and other hypogean habitats across the known distribution range of Troglohyphantes vignai and Pimoa rupicola in the Western Alps. The distribution range of T. vignai was entirely covered (including type locality and former localities of T. rupicapra, indicated hereinafter as T. vignai sensu rupicapra), while for Pimoa we only sampled Alpine populations, thus excluding French and Apenninic populations. The complete list of localities is reported in Table 1 and Fig. 1. The toponomastics and classification of the different sectors and sub-sectors of the Alps follows the partition of the Alpine chain (SOIUSA: Marazzi, 2005). Specimens were hand-collected, preserved in 95% ethanol and stored in freezer. Given that the sampled environments were highly oligotrophic, in certain localities we were able to detect and collect only few individuals of the two investigated species. The number of individuals collected for each locality ranged from 2 to 8 in P. rupicola and P. n. sp. and from 1 to 7 in T. vignai (Table 2). Overall, 119 Pimoa specimens from 25 localities and 37 Troglohyphantes specimens from 8 localities were used in this study.

Table 1 Summary of the sampled localities.

Sampled localities of Pimoa rupicola, P. n. sp. and Troglohyphantes vignai ordered by latitude (from North to South).

Cod	Valley	Locality	Habitat type	x	y	P. n.sp	P. rupicola	T. vignai	Date	Collector/s	
1	Susa	(!) Seinera	Abandoned mine	7,201	45,163	*			20.II.2011	Mammola S., Piano E., Giuliano D.	
2	Susa	(!) Dravejs	Scree	7,039	45,118	*			13.VI.2014	Mammola S., Piano E.	
3	Sangonetto	Coazze	Ruined building	7,241	45,067	*			20.II.2011	Isaia M.	
4	Sangonetto	Garida	Abandoned mine	7,304	45,055	*			20.II.2011	Isaia M.	
5	Chisone	[1591 Pi/TO] Tana del Diavolo	Wild cave	7,123	45,028	*		*	12.IX.2014	Isaia M., Mammola S.	
6	Chisone	Bocetto	Abandoned mine	7,086	44,959	*			12.IX.2014	Isaia M., Mammola S.	
7	Germanasca	[n.c. Pi/CN] Tuna du Diau	Wild cave	7,104	44,949	*		*	12.IX.2014	Isaia M., Mammola S.	
8	Lemina	S. Pietro Val Lemina	Abandoned mine	7,297	44,937	*			12.IX.2014	Isaia M., Mammola S.	
9	Germanasca	(!) Tornini	Abandoned mine	7,199	44,908	*		*	12.IX.2014	Isaia M., Mammola S.	
10	Germanasca	S. Germano Chisone	Abandoned mine	7,225	44,901	*			13.XI.2014	Isaia M.	
11	Pellice	[1538 Pi/TO] Gheisa d’la Tana	Wild cave	7,224	44,851	*			28.IX.2014	Isaia M., Mammola S., Paschetta M.	
12	Po	Balma di Rio Martino (Opera 372)	Military bunker	7,140	44,702	*			13.XI.2014	Isaia M., Mammola S., Paschetta M.	
13	Po	[1148 Pi/CN] Buco del Maestro	Wild cave	7,238	44,686	*			3.X.2014	Isaia M., Mammola S., Paschetta M.	
14	Po	[1009 Pi/CN] Buco di Valenza	Wild cave	7,172	44,683	*		*	13.XI.2014	Isaia M., Mammola S., Paschetta M.	
15	Varaita	(!) Tour Real	Blockhouse	6,982	44,645	*			29.VII.2014	Mammola S.	
16	Varaita	[1019 Pi/CN] Tana dell’Orso di Casteldelfino	Wild cave	7,099	44,561			*	21.VII.2013	Mammola S.	
17	Varaita	[1010 Pi/CN] Grotta di Rossana	Wild cave	7,431	44,534	*			20.VII.2013	Giresi A., Mammola S.	
18	Maira	[n.c. Pi/CN] Grotta del Partigiano di Roccabruna	Wild cave	7,294	44,509	*			14.VII.2014	Isaia M.	
19	Stura	[1122 Pi/CN] Grotta dello Scoiattolo	Wild cave	7,389	44,412	*			13.I.2015	Isaia M., Mammola S., Paschetta M.	
20	Stura	[1102 Pi/CN] Buco dell’ Aria Calda	Wild cave	7,462	44,349	*			03.X.2014	Isaia M., Mammola S., Paschetta M.	
21	Stura	[1056 Pi/CN] Grotta della Chiesa di Valloriate	Wild cave	7,382	44,339	*			13.I.2015	Isaia M., Mammola S., Paschetta M.	
22	Lisio	[884 Pi/CN] Grotta Rio dei Corvi	Wild cave	7,994	44,303	*			26.XII.2014	Isaia M., Mammola S.	
23	Corsaglia	[113 Pi/CN] Tana di Camplass	Wild cave	7,887	44,297		*		26.XII.2014	Isaia M., Mammola S.	
24	Vermenagna	Fort (B) of Vernante (Opera 14)	Military bunker	7,529	44,257		*		13.I.2015	Isaia M., Mammola S., Paschetta M.	
25	Pesio	[250 Pi/CN] Grotta superiore delle Camoscere	Wild cave	- Protected data -	44,21719			*	26.XII.2014	Isaia M., Mammola S.	
26	Tanaro	[118 Pi/CN] Grotta dell’Orso di Ponte di Nava	Wild cave	7,866	44,119		*		10.X.2014	Isaia M., Mammola S.	
27	Pesio	(!) Unknown cave near Colle del Pas	Wild cave	7,774	44,166			*	20.VIII.2014	Badino G.	
28	Argentina	[619 Li/IM] Sgarbu du ventu	Wild cave	7,937	44,002		*		27.XII.2014	Isaia M., Mammola S.	
29	Argentina	[104 Li/IM] Tana di Bertrand	Wild cave	7,867	43,916		*		27.XII.2014	Isaia M., Mammola S.	
Notes.

Cod, locality numeric code used in the analysis and figures. For each record we report the name of the locality, the name of the Alpine valley, the habitat type, the geographical coordinates (longitude and latitude in decimal degrees, WGS 84 reference system), the date and the collectors. For hypogean localities, we report the Speleological cadastrial number in square brackets (e.g., 1591 Pi/TO), when available. An exclamation mark in parenthesis (!) before the name of the locality indicates new unpublished records found during this study.

Table 2 Standard genetic diversity indices.

Diversity measures for the cox1 and ITS-2 genes for the localities of Pimoa n. sp., P. rupicola and Troglohyphantes vignai sampled in this study.

		Cox1			ITS-2			
Locality code	Species	N	H	π	h	N	H	π	h	
1	P. n. sp.	2	2	1,000	1,000	2	1	0,000	0,000	
2	P. n. sp.	2	2	1,000	1,000	2	1	0,000	0,000	
3	P. n. sp.	5	2	0,600	0,600	5	2	0,400	0,400	
4	P. n. sp.	4	1	0,000	0,000	2	1	0,000	0,000	
5	P. n. sp.	8	1	0,000	0,000	6	2	0,476	0,476	
6	P. n. sp.	6	1	0,000	0,000	6	1	0,000	0,000	
7	P. n. sp.	7	2	0,285	0,285	7	1	0,000	0,000	
8	P. n. sp.	3	3	0,000	0,000	3	2	0,666	0,666	
9	P. n. sp.	6	3	2,133	0,533	6	3	1,400	0,600	
10	P. n. sp.	2	1	0,000	0,000	2	1	0,000	0,000	
11	P. n. sp.	4	4	5,666	1,000	5	3	1,333	0,666	
12	P. n. sp.	7	4	2,476	0,714	8	3	0,678	0,464	
13	P. n. sp.	8	5	2,429	0,857	8	3	1,047	0,523	
14	P. n. sp.	5	2	1,500	0,500	5	2	0,333	0,333	
15	P. n. sp.	2	1	0,000	0,000	2	1	0,000	0,000	
17	P. n. sp.	5	2	2,500	0,500	4	1	0,000	0,000	
18	P. n. sp.	5	2	0,666	0,333	5	1	0,000	0,000	
19	P. n. sp.	3	3	2,666	0,666	3	2	0,000	0,000	
20	P. n. sp.	4	1	0,000	0,000	4	2	0,500	0,500	
21	P. n. sp.	5	1	0,000	0,000	5	1	0,000	0,000	
22	P. rupicola	2	1	0,000	0,000	2	1	0,000	0,000	
23	P. rupicola	5	1	0,000	0,000	5	1	0,000	0,000	
24	P. rupicola	4	1	0,000	0,000	5	1	0,000	0,000	
26	P. rupicola	6	1	0,000	0,000	6	1	0,000	0,000	
28	P. rupicola	7	1	0,000	0,000	7	5	2,285	0,857	
29	P. rupicola	3	2	7,333	0,667	3	1	0,000	0,000	
5	T. vignai	6	2	0,612	0,600	6	1	0,000	0,000	
6	T. vignai	6	2	1,000	0,500	6	3	1,166	0,833	
7	T. vignai	5	1	0,000	0,000	5	1	0,000	0,000	
9	T. vignai	7	3	0,571	0,523	7	1	0,000	0,000	
14	T. vignai	3	1	0,000	0,000	3	1	0,000	0,000	
16	T. vignai	3	1	0,000	0,000	3	1	0,000	0,000	
25	T. vignai *	6	3	0,666	0,600	6	1	0,000	0,000	
27	T. vignai *	1	1	0,000	0,000	1	1	0,000	0,000	
Notes.

N number of individuals

H number of haplotypes

π nucleotide diversity

h haplotype diversity

* Indicates populations of T. vignai sensu rupicapra.

DNA extraction, amplification and sequencing

One leg was removed from each specimen for DNA extraction. Whole genomic DNA was extracted from the samples using the SpeedTools Tissue Extraction Kit (Biotools) following the manufacturer’s protocol. A 676 bp region of the mitochondrial cytochrome oxidase subunit I (cox1) gene and a 400 bp region of the nuclear second internal transcribed spacer region (ITS-2) gene were amplified using polymerase chain reaction (PCR). We utilized the primers C1-J-1490 (5′-GGTCAACAAATCATAAAGATATTGG-3′; Folmer et al., 1994) and C1-N-2191 (5′-CCCGGTAAAATTAAAATATAAACTTC-3′; Simon et al., 1994) for the cox1 and the ITS-5.8s (5′-GGGACGATGAAGAACGGAGC-3′) and the ITS-28s (5′-TCCTCCGCTTATTGATATGC-3′) for the ITS-2 (White et al., 1990).

PCR amplifications were carried out in 25 µL reaction volume in a final concentration of 0.1 µL Taq polymerase (Promega), 5 µL buffer (Promega), 2.25 µL MgCl2 (Promega), 0.2 mm of each dNTP, 0.5 µL of each primer and 1.5 µL of DNA sample. PCR conditions for amplification were as follows: initial denaturing step at 95 °C for 5 min, 35 amplification cycles (94 °C for 30 s, 45 °C for 35 s, 72 °C for 45 s cox1 fragment and 94 °C for 45 s, 48 °C for 1 min, 72 °C for 60 s for ITS-2 fragment) fallowed by a final extension at 72 °C for 5 min. For certain populations of Pimoa rupicola (localities #23 and #26), a slightly different annealing protocol for the cox1 was utilized (94 °C for 30 s, 42 °C for 35 s, 72 °C for 45 s). PCR products were visualized on agarose gels.

PCR product were cycle-sequenced at Macrogen, Inc. (Seoul, Korea; http://www.macrogen.com). The DNA sequences obtained were preliminary assembled and edited using Geneious 7.1 (Kearse et al., 2012; http://www.geneious.com).

The alignment of the cox1 sequences was trivial, as they showed no evidence of indel mutations. The ITS-2 fragments were aligned with the online version of MAFFT (Katoh & Toh, 2008; http://mafft.cbrc.jp), using the Q-INS-I strategy with default options. We explored best partitioning schemes and substitution models simultaneously using PartitionFinder v.1.0.1 (Lanfear et al., 2012) under a Bayesian information criterion (BIC).

Genetic analyses

Population structure

Standard genetic diversity indices (nucleotide and haplotype diversity, A-T bias, transition/transversion rate) were estimated with the PopGenome package (Pfeifer et al., 2014) in R environment (R Development Core Team, 2013). We tested for population structure among localities in the cox1 dataset using FST as implemented in ARLEQUIN 3.01 (Excoffier, Laval & Schneider, 2005). Significance was assessed by performing 10,000 permutations. We excluded from this analysis six localities (#1, #2, #10, #15, #22 and #27) where the sampling representativeness was questionable (i.e., less than 3 individuals). Haplotype networks were constructed using the statistical parsimony method (Templeton, Crandall & Sing, 1992; Clement et al., 2002) with a confidence limit of 95% as implemented in PopArt (online at: http://popart.otago.ac.nz).

Phylogenetic inference

Maximum likelihood (ML) and Bayesian inference (BI) were used to infer the gene trees and the concatenated tree for each genus. For this analysis we only included unique haplotypes. We concatenated Cox1 and ITS-2 gene fragments in Geneious and excluded taxa with partial sequences. Gaps in the ITS-2 were recoded as absence/presence characters using the simple method proposed by Simmons & Ochoterema (2000) with the help of the computer program SeqState 1.4.1 (Müller, 2005). We used Troglohyphantes nigraerosae and Pimoa edenticulata as outgroups to root the respective trees based on the results of ongoing analyses on the two genera (MA Arnedo, 2015, unpublished data; G Hormiga, 2015, unpublished data).

ML analyses were performed in RAxML v.7.4.2 (Stamatakis, 2006) with the aid of the graphical interface RAXML-GUI v.1.3 (Silvestro & Michalak, 2011), by conducting 10 runs per 500 bootstrap replicates.

BI analyses were conducted in MrBayes v.3.2 (Ronquist et al., 2012) with two independent runs of 20 million generations with four Markov chains (one cold, three heated), sampling every 1,000 generations. The convergence of chains was checked in Tracer v.1.6 (Rambaut et al., 2014) until effective sample sizes (EES) was above 200, and the average standard deviation of split frequencies (ASDSF) of the two runs was below 0.02. The first 20% of trees in each run were discarded as burn-in. The majority-rule consensus tree was generated from remaining trees.

Divergence time estimation

Divergence time was estimated for the two lineages using a multispecies coalescent approach (Heled & Drummond, 2010), as implemented in BEAST (Drummond et al., 2012). Coalescent groups within each species were first identified by using the General Mixed Yule Coalescence (GMYC; Fujisawa & Barraclough, 2013) method and used as a proxy of species in the multicoalescent analyses. GMYC is a clustering method that provides an objective way to delimit putative independent evolutionary lineages (i.e., coalescent groups). For each cox1 alignment, we generated a ML tree (see analytical protocol above) and we converted it to an ultrametric tree with the help of PATHd8 (Britton et al., 2007). The GMYC analysis was conducted via the package splits (Ezard, Fujisawa & Barraclough, 2014) in R, after removing zero-length branches and make the tree fully dichotomous.

For estimating the divergence time of Troglohyphantes and Pimoa lineages, we utilized the best gene partition schemes estimated with PartitionFinder. Because of the lack of reliable calibration points (e.g., fossils, relevant geological or biogeographical events) for any of the two lineages, we relied on informed priors of the substitution rates of the cox1, based on available information for spiders (Bidegaray-Batista & Arnedo, 2011). Preliminary analyses using a lognormal relaxed clock for the cox1 gene showed that the posterior distribution of the ucld.mean parameter accreted to zero and hence a strict clock was preferred. We set the prior rate parameter of the cox1 strict clock to a normal distribution with mean ± sd = 0.02 ± 0.006. Similarly, we assigned a strict clock prior to the ITS-2 partition. To speed up calculation, we defined a flat prior to the ITS-2 mean rate parameter consisting in a uniform distribution with upper and lower bounds of 0.2 and 0.0001, respectively. We selected a Yule model for the tree prior.

For each species we ran three independent MCMC chains for 50 million generations, sampling every 10,000 generations. Convergence of the three chains and correct mixing was assessed in Tracer v.1.6 (Rambaut et al., 2014).

Ecological niche modeling

We relied on ecological niche modeling (ENM; see, e.g., Elith et al., 2006) to model the ancestral distribution of the target species. Detailed methodological protocol is provided in Supplemental Information 1. In a first step, we collected all records of the target species available in the literature. We managed to track down 22 localities for Troglohyphantes vignai, most of which clustered together. On the other hand, for Pimoa we recovered 110 localities (61 for Pimoa n. sp. and 49 for P. rupicola), including new unpublished records discovered during the present study. Given the low number of localities for T. vignai, we only inferred the ENM model for the Pimoa lineages. The dataset was corrected for potential spatial autocorrelation and haphazard sampling (Oliveira et al., 2014 and references therein). We obtained present day climatic data (19 ‘Bioclim variables’, see Table 1 of Supplemental Information 1) and altitude a.s.l. from the WorldClim website (www.worldclim.org). We obtained downscaled and calibrated Paleoclimatic data for the Last Glacial Maximum (∼22,000 years ago; hereinafter LGM) from three different simulations available from Global Climate Models (GCMs; Coupled Model Intercomparison Project phase 5; http://cmip-pcmdi.llnl.gov/cmip5). The climatic preferences of the two species were investigated via Principal Component Analysis (PCA) in the Vegan R package (Oksanen et al., 2013). We investigated collinearity among covariates and obtained a final set of uncorrelated variables (Annual mean temperature (Bio1), Temperature annual range (Bio7) and Mean temperature of the driest quarter (Bio9)).

We generated presence-only models with the Maximum Entropy Distribution Models available in MaxEnt (Phillips, Anderson & Schapire, 2006), as implemented in the dismo R package (Hijmans et al., 2014). Firstly, we computed the models on the present climate and on the occurrence points within the M area (sensu Barve et al., 2011; details in Supplemental Information 1). To generate the prediction, we ran each niche model 20 times using a loop script in R, keeping in all cases a random partition of 20% of the occurrence points, which was used to evaluated model performance via the Area Under the Curve (AUC) of the Receiver Operating Characteristic (ROC) plot (Fielding & Bell, 1997). We projected the MaxEnt models in the LGM climate, under each of the three GCMs climatic scenarios. We used a conservative approach to identify potential Pleistocene refugia: we first applied a threshold of 0.6 to the continuous probability surface of presence estimated after the projection. We then combined the three projections, by sub-sampling only those pixel classified as potentially occupied (p > 0.6) in each LGM forecast.

Results

Population structure

The new sequences obtained in the present study are available in GenBank (Supplemental Information 2). A fragment of the mitochondrial cox1 gene of 676 bp was obtained for 37 specimens of Troglohyphantes vignai in 8 localities, corresponding to 14 unique haplotypes. The cox1 data set had 79 segregating sites and 9 parsimony informative sites. The overall mean distance (p-distance among haplotypes) was 0.0495 ± 0.0059. Sequences of the nuclear intron ITS-2 were obtained from the same individuals. The alignment was 400 positions long, 10 additional absence/presence gap characters were scored, corresponding to 10 ITS-2 sequence types. The ITS-2 had 16 segregating sites and 9 parsimony informative sites. The overall p-distance among caves was 0.0389 ± 0.0020. We obtained 676 bp cox1 sequence fragments of 119 Pimoa individuals from 25 localities. The 93 individuals from 19 localities of Pimoa n. sp. yielded 43 haplotypes (35 segregating sites and 7 parsimony informative sites) and the 27 individuals from 6 localities of P. rupicola yielded 7 haplotypes. The average p-distance within populations in Pimoa n. sp. and P. rupicola was 0.0076 ± 0.0017 and 0.0052 ± 0.0017, respectively, and the maximum p-distance between the two lineages was 0.1164 ± 0.0111. The nuclear ITS-2 sequences were obtained from 118 Pimoa specimens. The final alignment included 411 positions and 4 additional gap characters. Individuals of Pimoa n. sp. (90 individuals) and P. rupicola (28 individuals) yielded 34 and 10 sequence types, respectively. The average p-distance within populations in Pimoa n. sp. and P. rupicola was 0.0102 ± 0.0027 and 0.0035 ± 0.0017, respectively, and the maximum p-distance between the two lineages was 0.0701 ± 0.0121.

The standard genetic diversity indices calculated for the cox1 and the ITS-2 for each locality are summarized in Table 2. Pimoa n. sp. showed high levels of nucleotide diversity and low levels of haplotype diversity in most of the populations, while both P. rupicola and T. vignai showed low levels of haplotype and nucleotide diversity. This pattern was especially obvious in P. rupicola, since most individuals within populations were identical.

Pairwise FST values calculated for the localities of the three species are reported in Table 3. Pairwise FST values between localities revealed contrasting patterns of gene flow. In T. vignai, pairwise FST values between localities were always higher than 0.8, and significant (p < 0.05), except for localities #14 and #16. A relatively strong population structure was also found in P. rupicola. Pairwise FST values between localities were always higher than 0.6, although significant comparisons involved exclusively southernmost localities (#28 and #29). Pimoa n. sp. showed instead a more shallow population structure, with several FST values below 0.5, generally corresponding to nearby localities.

Table 3 Population structure among localities.

FST values for mtDNA cox1 of Pimoa n. sp., P. rupicola and Troglohyphantes vignai based on the Tamura and Nei model. Locality codes are explained in Table 1. Localities #1, #2, #10, #15, #22 and #27 were excluded from the analysis, being represented by less than three individuals.

FSTPimoa n. sp.	
	3	4	5	6	7	8	9	11	12	13	14	17	18	19	20	21	
3	0,000	–	–	–	–	–	–	–	–	–	–	–	–	–	–	–	
4	0,250	0,000	–	–	–	–	–	–	–	–	–	–	–	–	–	–	
5	0,723	0,018	0,000	–	–	–	–	–	–	–	–	–	–	–	–	–	
6	0,787	0,123	0,332	0,000	–	–	–	–	–	–	–	–	–	–	–	–	
7	0,557	0,322	0,412	0,341	0,000	–	–	–	–	–	–	–	–	–	–	–	
8	0,700	0,764	0,597	0,422	0,857	0,000	–	–	–	–	–	–	–	–	–	–	
9	0,433	1,000	0,733	0,733	0,590	0,733	0,000	–	–	–	–	–	–	–	–	–	
11	0,200	1,000	0,500	0,500	0,357	0,500	0,230	0,000	–	–	–	–	–	–	–	–	
12	0,340	0,561	0,642	0,642	0,500	0,642	0,376	0,142	0,000	–	–	–	–	–	–	–	
13	0,289	0,642	0,589	0,578	0,446	0,589	0,322	0,006	0,218	0,000	–	–	–	–	–	–	
14	0,500	0,589	0,800	0,800	0,657	0,800	0,533	0,300	0,442	0,357	0,000	–	–	–	–	–	
17	0,450	1,000	0,750	0,750	0,607	0,750	0,483	0,250	0,392	0,186	0,470	0,000	–	–	–	–	
18	0,533	0,750	0,833	0,830	0,690	0,833	0,566	0,333	0,476	0,422	0,633	0,583	0,000	–	–	–	
19	0,366	0,860	0,660	0,666	0,523	0,666	0,400	−0,071	0,309	0,107	0,466	0,416	0,500	0,000	–	–	
20	0,700	1,000	1,000	1,000	0,857	1,000	0,733	0,250	0,642	0,452	0,800	0,750	0,833	0,000	0,000	–	
21	0,800	1,000	1,000	1,000	0,857	1,000	0,733	0,500	0,642	0,452	0,750	0,000	0,833	0,666	1,000	0,000	
FSTPimoa rupicola		FSTTroglohyphantes vignai			
	23	24	26	28	29			5	6	7	9	14	16	25			
23	0,000	–	–	–	–		5	0,000	–	–	–	–	–	–			
24	1,000	0,000	–	–	–		6	0,916	0,000	–	–	–	–	–			
26	1,000	1,000	0,000	–	–		7	0,964	0,855	0,000	–	–	–	–			
28	1,000	0,667	1,000	0,000	–		9	0,957	0,873	0,943	0,000	–	–	–			
29	0,876	0,876	0,876	0,667	0,000		14	0,991	0,988	1,000	0,991	0,000	–	–			
							16	0,989	0,984	1,000	0,989	1,000	0,000	–			
							25	0,987	0,984	0,992	0,988	0,975	0,988	0,000			
Notes.

Values in bold represent significant comparisons (p < 0.05).

Pimoa haplotypes were resolved as two independent networks, corresponding to Pimoa n. sp. (1) and P. rupicola (2), respectively, separated by 42 steps. P. rupicola haplotypes were limited to single populations, except H26 and H27, which were shared across populations. Generally, populations had low haplotype diversity, and in one case (#26) there was one single haplotype. Two divergent haplotypes (11 steps), however, were found in locality #29.

In Pimoa n. sp. haplotypes from nearby localities clustered together or were separated by only few steps (1–4). Several haplotypes (H7, H8, H12, H14) were shared among closely located localities (e.g., occurring in the same Alpine valley or in adjacent valleys). In some instances, however, haplotypes from distant localities were found to be very similar (few steps), e.g., locality #1 and #2 with #12. Moreover, two haplotypes (H18 and H22) were found in individuals occurring in distant populations.

In T. vignai, haplotypes were not shared between localities. Haplotypes from closely located populations were generally separated by few mutations. Haplotype diversity was low within populations, and localities #7, #14 and #16 showed single haplotypes. Two localities (#9 and #25), on the other hand, had more than two haplotypes. The haplotypes of T. vignai sensu rupicapra were separated by 17–20 steps from the nearest locality (#14), which is actually the locus typicus of T. vignai (Buco di Valenza cave). A higher number of steps (38–56) separated this cave from the remaining localities.

Phylogenetic tree and estimation of the divergent time

Partition Finder selected the full codon as the best partition scheme for the alignments of both species. The models for each partition are reported in Table 2 of Supplemental Information 1. Both the Bayesian and ML analyses of the concatenated data matrix of T. vignai yielded in similar tree topologies, and most branches were highly supported (i.e., posterior probabilities (PP) > 0.95, bootstrap support (BS) < 75%; Fig. 2). T. vignai was split in two main clades: one including the southern populations (#14, #16, #25 and #27), and the second one including the remaining northern populations (#5, #6, #7 and #9). The Bayesian and the ML analyses also recovered similar tree topologies in Pimoa (Fig. 3). Two well-supported clades were detected, corresponding to Pimoa n. sp. and P. rupicola, respectively. Individuals from geographical adjacent localities were closely related, although basal branches within Pimoa n. sp. were poorly supported.

Figure 1 Haplotype networks of the investigated populations.

Statistical parsimony haplonetworks for Pimoa n. sp. (A), P. rupicola (B) and Trogolohyphantes vignai (C). Numbers in maps indicate localities (see legend), alphanumeric codes in the networks refer to haplotypes. The size of each circle is proportional to the number of sampled individuals with each haplotype (see scale above the legend). Unsampled and/or extinct haplotypes are represented by small black circles.

Figure 2 Phylogenetic tree of Troglohyphantes.

Topology obtained in the concatenated Bayesian analysis for Troglohyphantes vignai. Only one individual per haplotype is shown. Vertical rectangles denote support as follows: Bayesian posterior probabilities (PP; left rectangles) and maximum likelihood bootstraps (ML; right rectangles); black: PP > 0.95, ML bootstrap support > 70%, white: support values lower than threshold values. The asterisk (*) indicate the locus typicus of T. vignai. Localities #25 and #27 refer to T. vignai sensu rupicapra.

Figure 3 Phylogenetic tree of Pimoa.

Topology obtained in the concatenated Bayesian analysis for Pimoa. Only one individual per haplotype is shown. Vertical rectangles denote support as follows: Bayesian posterior probabilities (PP; left rectangles) and maximum likelihood bootstraps (ML; right rectangles); black: PP > 0.95, ML bootstrap support > 70%, white: support values lower than threshold values.

The GMYC algorithm identified 2 coalescent clusters within Pimoa cox1 sequences (ML = 100.8932; LR = 28.59767; p < 0.000), one including all sequences/localities of P. rupicola and the other including all sequences/localities of Pimoa n. sp. Troglohyphantes vignai cox1 sequences were resolved as 7 coalescent clusters (ML = 424.2824; LR = 207.4615; p < 0.000; Fig. 4). Except for cluster A2, which included individuals from two caves, each cluster corresponded to individuals sampled from single caves.

The species trees and the embedded cox1 gene tree recovered for each spider genus are shown in Fig. 4. The substitution rate estimated for the Troglohyphantes cox1 was 0.0218 substitutions per lineage/million years (95% HPD = 0.010–0.033), and for the ITS-2 was 0.0024 (95% HPD = 0.0005–0.0031). The split between T. vignai and T. nigraerosae was traced back to approximately 7.2 million years ago (Ma, 95% Highest Posterior Density, HPD = 13.7–3.5 Ma). Diversification of the extant T. vignai lineages occurred 2.9 million years ago (95% HPD = 5.4–1.5 Ma), while the diversification of the extant northern populations (D2, E2, F2, G2 clusters) occurred approximately 0.5 Ma (95% HPD = 0.9–0.2 Ma). The substitution rate estimated for Pimoa cox1 was 0.0217 substitutions per lineage/million years (95% HPD = 0.011–0.033), and for the ITS-2 was 0.006 (95% HPD = 0.002–0.013). The basal split between Pimoa n. sp. and P. rupicola was estimated to have occurred around 5.7 Ma (HPD = 12–2 Ma). The origin of the extant diversity of each lineage was estimated approximately at 0.4 Ma for both lineages (HPD = 1–0.15 and HPD = 0.85–0.15 Ma, for P. rupicola and Pimoa n. sp. respectively).

Figure 4 Timeframe of diversification.

Chronograms obtained with the multispecies coalescent approach for the cox1 and ITS2 genes combined (orange topologies) and the cox1 gene alone (black topologies). Grey node bars indicate the 95% HPD confidence intervals of the divergence time (for sake of clarity, only those HPD referring to the cox1 gene are shown). The common x-axis is time in million years (Mya).

Climatic segregation

We studied the climatic preference for the two Pimoa linages with PCA, using bioclimatic variables (see Table 1 of Supplemental Information 1). The bi-plot of scores for the first two axes of the PCA is shown in Fig. 5. The first two axes explained 86.2% of the variation in the data. The variance explained by other axis was negligible (<1% each). In respect to the first two axes, the two species segregate into two distinct clusters. The first axis (eigenvalues = 12.28; variance explained = 62.0%) mostly reflect a gradient of temperature. The second axis (eigenvalues = 4.79; variance explained = 24.2%) was positively correlated with variables reflecting diurnal and annual thermic excursion (Bio2, Bio4, Bio7) and anticorrelated with variables reflecting seasonal precipitation (Bio13, Bio16, Bio19). Although the first axis (PC1) explains most of the variance in the data, the localities of the two Pimoa species cluster in two groups according to the second axis (PC2), which combines bioclimatic variables referable to continentality. According to the original definition (see Currey, 1974 for more details), continentality is intended as a measure of how the climate is affected by its remoteness from the sea. Specifically, the distance from water masses influences the climate in terms of higher seasonality (Bio4), increasing diurnal (Bio2) and annual (Bio7) temperature ranges, as well as decreasing precipitation in the coldest (Bio19) and wettest (Bio13 and 16) periods. In light of our results, Pimoa n. sp. occurs in areas characterized by higher continentality in respect to P. rupicola.

Figure 5 Climatic segregation of the Pimoa lineages.

Bi-plot of Principal Component Analysis (PCA) scores for the first two axes based on 19 bioclimatic variables and altitude a.s.l. extracted for the localities of Pimoa n. sp. (purple dots) and Pimoa rupicola (green dots). For the explanation of the bioclimatic variables see Table 1 in Supplemental Information 1.

Current and past distribution of Alpine Pimoa

The predictive performance of our bioclimatic models was fairly high both in Pimoa n. sp. (mean ± SD AUC of the 20 runs = 0.845 ± 0.053) and in P. rupicola (0.908 ± 0.089). Overall, the suitable areas predicted by the model were congruent with the known distribution of the two species, at least in the Western Alps and Apennines. Current predictions identified suitable areas for P. n. sp. around the medium mountain belt (500–1500 m a.s.l.) of the Central and Western Alps, from the Camonica valley (province of Bergamo) down to the margin between Cottian and Maritime Alps (Stura valley, province of Cuneo). In respect to the known distribution, the predicted range of P. n. sp. extended northwards over the known limit of the species (see dotted line in Fig. 6A). More suitable areas were also detected in the northern edge of the Tuscan Apennines (Fig. 6A). The suitable areas of P. rupicola corresponded to the southern border of the Western Alps, in the coastal belt that spreads from Côte d’Azur (SW France) to the Ligurian eastern coast and Tuscan Apennines (Italy). Additional suitable areas were also found in Tuscany, Lazio and Corsica. The projection of the distribution model to the environmental condition of the Last Glacial Maximum (LGM) revealed that most of the current suitable areas were unsuitable in the LGM (Fig. 6B). Our threshold approach identified one main refugia for P. n. sp. (RF1) and two for P. rupicola (RF2 and RF3). All refugia corresponded to areas that were devoid from glaciers (in accordance with Ehlers, Gibbard & Hughes, 2011). The RF1 extended outside the southern edge of the actual distribution of Pimoa n. sp., in the hills and plains surrounding roughly the Northern border of Maritime and Ligurian Alps. RF2 corresponded to small areas along the French and Italian Riviera. RF3 extended over a wider geographic area in the Apennine, and in the northern part of the Corsica.

Figure 6 Current and past distribution of Pimoa lineages.

Maps of the predicted environmental suitability according to the ENMs fitted to the occurrence points for Pimoa n. sp. (purple surface) and P. rupicola (green surface), at the present climate (A) and during the Last Glacial Maximum (B). Potential Pleistocene refugia (RF1, RF2, RF3) were identified by combining the three GCMs climatic reconstructions and applying a threshold of 0.6. The northern limit of the known distribution of P. n. sp., corresponing to the Graian Alps (Isaia et al., 2011), is highlighted in the upper map with a dotted line. Limits of the ice cover in the Last Glacial Maximum (Ehlers, Gibbard & Hughes, 2011) are reported for Pleistocene projections (white shapes in the lower map).

Discussion

The history of two cave-dwelling spiders

The confounding effects of adaptation, biogeography and dispersal ability on the origin and the distribution patterns of cave organisms pose a stimulating challenge to biogeographers (Porter, 2007; Juan et al., 2010). According to Culver & Pipan (2010), long term climatic changes can be claimed as the main factors that prompted invertebrate species to colonize the subterranean habitats. In this regard, the Miocene climatic transitions and the Pleistocene glaciations are considered among the most important events. Here, by reconciling phylogeographic patterns and predictive ENMs, we provide support to this view, pointing out the Cenozoic climatic transitions has the most important factors shaping the present day genetic diversity and the distribution range of our model species.

Although special caution should be exercised when considering time estimates based on molecular data, especially in the absence of fossil record (Hipsley, 2014), our results fits well with some of the major climatic event undergone in the Western Alps during the Cenozoic. Accordingly, the two Pimoa lineages and Troglohyphantes vignai originated from the Middle (Serravallian) to the Late Miocene (Messinian), namely from 13 to 3.5 mya. More precisely, the isolation of T. vignai and P. rupicola from their northern sister groups (T. nigraerosae and P. n. sp.) dates back 7.2 and 5.7 mya, respectively. This time period approximately corresponds to the closure of the Gibraltar Strait and the onset of the so-called Messinian Salinity Crisis (MSC; after Ruggieri, Adams & Ager, 1967). However, given the large confidence intervals around our time estimates (∼10 million years), it is difficult to draw precise conclusions about the event that exactly determined the split of the different lineages.

It is worth noticing that the onset of a climatic transition in the Middle Miocene, marked the decline of the last global climate optimum conditions, leading to a progressive deterioration of the dominant subtropical climatic conditions (Suc, 1984; Shevenell, Kennett & Lea, 2004; Jiménez-Moreno, Fauquette & Suc, 2010). It is arguable that in parallel to the slow climate deterioration and the increase of seasonality, isolation of Pimoa and Troglohyphantes occurred.

Being possibly pre-adapted to shallow moist humid habitats (Deeleman-Reinhold, 1978; Hormiga, 1994; Zhang & Li, 2013; Wang et al., 2008), both species progressively colonized the subterranean habitat, most likely during the Pleistocene. Given their contrasting level of troglomorphism, it is likely that the process began earlier in Troglohyphantes.

The known distribution range of T. vignai stretches discontinuously along the Cottian (Chisone, Po and Varaita valleys) down to Ligurian (Valle Pesio) Alps. Conversely, the distribution of the northern sister group T. nigraerosae is adjacent southern Graian Alps (Isaia et al., 2011). The lack of shared haplotypes and the FST values close to 1 between the sampled localities of T. vignai indicates a strong isolation of the populations (Holsinger & Weir, 2009). The same idea is further corroborated by the identification of each population as an independent coalescent lineage (i.e., GYMC cluster). These results are in agreement with other studies on subterranean arachnids (Hedin, 1997; Hedin & Thomas, 2010; Dixon & Zigler, 2011) that support the “caves as islands” scenario (sensu Snowman, Zigler & Hedin, 2010), in which dispersal is virtually absent and the different populations diverge in allopatry. Under such conditions, it seems likely the subterranean habitat acted as an evolutionary cul-de-sac (see Fišer, Blejec & Trontelj, 2012) for T. vignai. According to our time estimates the diversification of extant T. vignai lineages (especially northern population) occurred approximately during the Pleistocene glaciations. In this respect, it is worth noting that subterranean localities inhabited by T. vignai lie at the periphery of the Pleistocene glaciers (Ehlers, Gibbard & Hughes, 2011; see also local glacial limits reconstructed in Motta, 2014). Because subterranean populations most likely cannot survive under the ice cover (Culver & Pipan, 2010), we suggest that the present day distribution range of T. vignai is the shadow of a wider ancestral distribution. Populations inhabiting the northern valleys in the Cottian Alps (Germanasca and Chisone valleys; see Fig. 4), where the ice shield was more compact, provide further evidences of the effect of Quaternary ice sheets.

In this area the distribution of T. vignai overlaps with the range of several hypogean species of Doderotrechus beetles (Carabidae, Trechini) (Giachino, 1993; Giachino & Vailati, 1997; Casale & Giachino, 2008). The similarity of distribution of both groups is most likely the result of a similar response to glacial dynamics. Gaining further knowledge on the biogeography of other subterranean species may provide further confirmation for the patterns here recovered.

In contrast with T. vignai, for which we did not detect any evidence of current population expansion, the topology of the chronogram obtained for Pimoa lineages hints at a recent expansion following a bottleneck (see Fig. 4). The putative population expansion in both lineages of Pimoa would fall within the Quaternary Glacial Cycles, between 2.8 and 2.5 Ma (Gibbard, Head & Walker, 2010). The movement of glaciers as well as the continuous formation and melting of new ice sheets may have deeply affected the different populations, altering profoundly the local habitat suitability. Such transformations prompted either the migration of populations to more suitable areas at lower altitudes or latitudes or the local extinction of resident populations. Such scenario is congruent with the genetic fingerprint found in both Pimoa lineages. Accordingly, we hypothesize a glacial cycle-driven extinction of ancestral populations during cooler periods, followed by the expansion of populations which survived in climatic refugia during warmer periods. A similar pattern was observed in the Asian species P. clavata (Wang et al., 2008). The ENMs projected into the paleoclimatic reconstruction pointed out three putative areas devoid from glaciers that may have acted as glacial refugia for the surviving populations of each lineage (RF1-3; Fig. 6B). In the case of P. n. sp., we detected one main macrorefugia (RF1) associated to the southernmost offshoots of the Alpine glacial masses. Notably, similar areas have also been classified as peripheral refugia for several Alpine plants, such as Phyteuma globulariifolium (Schönswetter et al., 2002) and Ranunculus glacialis (Schönswetter et al., 2003). For P. rupicola, potential macrorefugia where located along the SW French coast (RF2) and the Tuscan coast (RF3), including the northern part of Corsica. Even though the presence of P. rupicola in Tuscany is confirmed by literature records (Brignoli, 1971; Hormiga, 1994), recent investigation conducted by the authors did not confirm the present occurrence of the species in Corsica. Because of the larger spatial resolution of the LGM stacked rasters compared to the present day data (2.5 min versus 30 arc-seconds (∼1 km)), it should be borne in mind that we may have not detected small point-like microrefugia (sensu Rull, 2009) within the interior of the Pleistocene ice shield covering the Alps.

As suggested by the lower FST values compared to Troglohyphantes, the relatively fast recolonization of ice-free areas is probably the result of the more effective dispersal ability of Pimoa. In particular, the population expansion followed a south-north direction, leading to the present distribution ranges of both lineages. Concerning P. n. sp., we additionally predict suitable areas up to the Central Alps. Given the continuity of the suitable habitats predicted by the ENM and the supposed high dispersal ability of Pimoa, we hypothesize an ongoing expansion of the populations northwards. Indeed, the occurrence of Pimoa in outer shaded and humid habitats such as beech forests and other broadleaved forests (Bertkau, 1890; Jackson, 1926; Thaler, 1976; Isaia et al., 2015; Isaia, Paschetta & Chiarle, 2015) provides empirical evidence of the existence of epigean dispersal. Because of the sex bias among the specimens collected in superficial habitats (Isaia et al., 2015; Isaia, Paschetta & Chiarle, 2015) and the general trend observed for spiders (Foelix, 1996), gene flow appears mostly mediated by males.

It should be borne in mind, however, that in light of the relatively low sample-size and the potential bias linked to the possible male-mediated dispersal, our cox1 data (and the associated FST results) may not fully reflect patterns of gene flow (see, e.g., Willing, Dreyer & Van Oosterhout, 2012; Davalos & Russell, 2014). Since low sample size is known to impact this kind of estimations, caution should be exercised when interpreting the values of nucleotide and haplotype diversity calculated for each locality (e.g., Goodall-Copestake, Tarling & Murphy, 2012). To minimize these potential bias, we left out from the calculation of the FST populations consisting of less than three individuals. Nevertheless, it is worth noticing that during data exploration we obtained very similar patterns of gene flow when comparing cox1 and ITS-2 results.

At present, the two lineages of Pimoa identified in this study show allopatric distributions (Fig. 6). P. n. sp. populations occur preferentially in areas characterized by higher continentality, and seem to tolerate cooler temperatures at higher altitudes and latitudes, as suggested by the results of the PCA (Fig. 5). On the other hand, Pimoa rupicola occurs in less continental areas, characterized by relatively small seasonal variations and high mean annual temperatures (i.e., Mediterranean climate). Similar complete niche partitioning between congeneric subterranean spiders has been reported elsewhere (Ribera, 1978; Gasparo & Thaler, 2000; Mammola & Isaia, 2014).

The application of ENM techniques has become a widespread practice to answer biogeographical and evolutionary questions (Franklin, 2009). In particular, ENM have been extensively used to identify Pleistocene refugia (e.g., Waltari et al., 2007; Rodriguez-Sanchez & Arroyo, 2008; Peterson, 2009; Planas et al., 2014). In constructing our scenarios, we have adopted a conservative approach, as our goal was to generate predictions under different levels of uncertainty. Although we relied on this approach, we are aware that ENMs have been rarely—and only recently—applied to study the hypogean ecosystems (see, e.g., Bryson et al., 2014; Camp et al., 2014; Naranjo, Moreno & Martiín, 2014). This is probably because, in first approximation, the link between the climatic variables (i.e., the external climate) and the subterranean habitat is not so straightforward. However, temperature of the underground compartment generally reflects the climatic regimen on the surface (Smithson, 1991; Badino, 2010). Although less intuitive, the regimen of rainfall plays an equally crucial role—if not more important—in determining such conditions (see details in Badino (2004) and Badino (2010)).

Overlooked diversity

In light of our results, some consideration regarding the overlooked diversity of our model species can be drawn. Concerning T. vignai, in this study we have included specimens from eight different localities, including topotypical material and material of T. vignai sensu rupicapra. The low levels of genetic variability observed between the latter and the topotypic material of T. vignai (Fig. 2, p-distance = 0.0022), provide further support for the synonymy between the two species proposed by Pesarini (2001). Therefore specimens of T. vignai sensu rupicapra have to be regarded as a population of T. vignai isolated in an area characterized by different climatic conditions (major Mediterranean influence). Such isolation could tentatively be related to the higher development of troglomorphism in T. vignai sensu rupicapra, as already observed by Brignoli (1971) and Isaia & Pantini (2010). Moreover, climatic factors provide a further line of interpretation for the presence of two main lineages within T. vignai, corresponding to the northern and the southern clade (Figs. 1 and 2). Slight differences in the shape of the lamella significativa of the male palp could lead to consider the two lineage as candidate species (Vieites et al., 2009), however the genetic distance between the two lineages, is about half of the value observed in the two nominal species T. vignai and T. nigraerosae.

The genus Pimoa is represented worldwide by 28 species (World Spider Catalog, 2015), many of which have only been described recently (Xu & Li, 2007; Xu & Li, 2009; Trotta, 2009; Hormiga & Lew, 2014). The application of molecular tools to investigate fine scale phylogeographic patterns in this group may uncover additional hidden diversity (Wang et al., 2008). It is generally accepted that species delimitation and eventually species description should be based on the integration of multiple lines of evidence (Padial et al., 2010). Here, we uncovered two deeply divergent genetic lineages (Fig. 3, GYMC clusters with p-distance above 9%) within Pimoa rupicola, which are further delimited both by genitalic morphology (S Mammola, G Hormiga, MA Arnedo, M Isaia, 2015, unpublished data) and different ecological requirements (Fig. 5).

Conclusions

Here, we have described the origin and the subsequent diversification of two species of spiders with contrasting levels of troglomorphism. We suggest that a different level of adaptation to subterranean life is an important factor to consider in the study of phylogeographic patterns. In particular, the major climatic events that occurred in the Alps during the Cenozoic determined from one side the complete isolation of pre-adapted subterranean species causing present day high population structuring, and from the other, the obliteration of surface-dwelling populations, causing their extinction or the lack of genetic structure in present day populations.

The parallel study of populations of subterranean species, especially when showing different levels of adaptation and overlapping ranges of distribution, may disclose new ways to understand patterns of biological diversification. Future research may include new highly variable nuclear markers and analytical tools, and also consider other taxa showing similar distributions (e.g., Doderotrechus beetles), to shed further light on the processes that shaped the present diversity of Alpine subterranean fauna.

Supplemental Information

Supplemental Information 1 Supplementary materials 1

Click here for additional data file.

Supplemental Information 2 Supplementary materials 2

Click here for additional data file.

We are indebted to Eli Mora, Leticia Bidegaray-Batista, Ana Riesgo, Sergio Taboada and Vanina Tonzo for their useful suggestions on the analysis and the interpretation of the results. SM would especially thank Paola Mazzuca for guiding him through the molecular lab protocols. We are grateful to Carola Ponzetto for proofreading a preliminary version of this manuscript and to Irene Frigo for the help in creating the layout of Figs. 1–4. Finally, a very special thanks goes to all the friends and colleagues that helped us in the collecting (Mauro Paschetta, Elena Piano, Davide Giuliano, Alessandro Giresi and Giovanni Badino) and to Gustavo Hormiga for providing material of Pimoa edenticulata for DNA comparison.

Additional Information and Declarations

Competing Interests

Author Contributions

DNA Deposition

The authors declare there are no competing interests.

Stefano Mammola conceived and designed the experiments, performed the experiments, analyzed the data, wrote the paper, prepared figures and/or tables, sampled hypogean habitats.

Marco Isaia conceived and designed the experiments, reviewed drafts of the paper, sampled hypogean habitats.

Miquel A. Arnedo conceived and designed the experiments, analyzed the data, contributed reagents/materials/analysis tools, reviewed drafts of the paper.

The following information was supplied regarding the deposition of DNA sequences:

GenBank: Accession numbers KT832079–KT832388.

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
