# Peer review of "Alpine endemic spiders shed light on the origin and evolution of subterranean species"

_PeerJ, doi:10.7717/peerj.1384_

## Round 0.1 · original submission · Major Revisions

· Academic Editor

Major Revisions

We received two really quality reviews of your paper. I think you can deal with the reviewers concerns but it will take some time and effort, especially with respect to reviewer 2 who rightfully questions your conclusion that climatic changes associated (with) the Messinian Salinity Crisis were the isolating force in these spiders. Be sure you include a detailed response to each review concern when you submit your revision. I will carefully evaluate how you dealt with each concern.

Reviewer 1 ·

Basic reporting

-Introduction needs to include a sentence or two explaining the validity of the two genes you chose.
- Low sample sizes are known to impact estimations of nucleotide and haplotype diversity (e.g. W P Goodall-Copestake, G A Tarling and E J Murphy. On the comparison of population-level estimates of haplotype and nucleotide diversity: a case study using the gene cox1 in animals. Heredity (2012) 109, 50–56). Concerns should be included in the discussion and mention of the sample sizes themselves should have been reported earlier in the analysis or methodology rather than exclusively in Table 2.
-Likewise, the mention of male mediated dispersal should be included in the introduction as it is a potential bias.
-Manuscript has several grammatical errors and typos – needs to be proofed.
-Would you suggest considering adaptive gene markers along with, as you did, inferring modes of selection?
-How is your study relevant on a broader scale?

Experimental design

-Part of this analysis uses mitochondrial DNA in organisms with male-mediated dispersal. Could this have led to an overestimate of population differences in any analyses depending on the cox1 marker (e.g. FST in lines 184-187)? Could use of mtDNA in estimation of divergence time have overestimated time since divergence? Did you formally compare conclusions from cox1 to conclusions using ITS-2? This should be reported. See Davalos and Russell (2014) Sex-biased dispersal produces high error rates in mitochondrial distance-based and tree-based species delimitation. Journal of Mammalogy,
95(4):781–791, 2014

-Line 100 mentions: “Given the higher degree of cave specialization, we further hypothesized that the genetic diversity within caves and the gene flow between caves would be very limited in T. vignai.” – could you really evaluate diversity within caves when sample sizes were so low?

-How could you address the variability in diversity metrics computed within caves using the two different markers? (Table 2) Is this because of small sample sizes?

Validity of the findings

The discussion still seems to reflect take-home messages of the analytical results that might not change by considering potential biases of mtDNA and low sample sizes. However, the complications with reporting these estimates need to be better expounded, along with an explanation of how overall conclusions might differ in the case of either bias.

Reviewer 2 ·

Basic reporting

22 Delete ‘several’

Abstract in general. You might instead spend some effort in the Abstract noting the different population genetic patterns observed (e.g. greater population structure in Troglohyphantes) between the species and how that relates to levels of troglomorphism.

45 ‘Long term’, not ‘Long terms’

79 “1009 Pi/CN” These abbreviations (here and elsewhere) need to be defined or clarified.

150 Why use ‘leg’ instead of ‘collector’ or something similar?

Table 1. What is a ‘Crambling’ house?

Table 1. Is it appropriate to reveal cave locations at such high resolution? I see one is marked as protected data so perhaps the other locations are in the public domain already.

200 ‘incomplete taxa’ should perhaps be ‘taxa with partial sequences’

238 ’million’ not ‘millions’

239 ‘in’ not ‘into’

279 ‘parsimony informative sites’ not ‘parsimony sites’

292-293 too many significant digits.

306-307 why do the p-values have asterisks?

347 ‘limited to’ instead of ‘exclusive of’

Figures 1 & 6 the background of the maps is too dark. It makes it hard to distinguish the sample sites and their colors.

Figure 3. Is the scale bar incorrect? 0.09 seems incorrect. Should this be 0.009 as in Figure 2?

425. ‘Concerning PCA results,’ is an awkward start to this section. Revise.

433. Clarify ‘gradient of continentality’

506. Perhaps ‘The history of two cave-dwelling spiders’

580-582. Expand to note this means your COI data (and Fst results from those data) may not fully reflect patterns of gene flow.

658. why is ‘taxa’ italicized?

Experimental design

305-309 Tajima’s D assumes a panmicitic population which is far from the case here. Lack of panmixia can influence these results significantly (e.g. Stadler et al. 2009 in Genetics). This needs to be noted as a major caveat or the Tajima D results should be left out. Conclusions based on these results (on population changes) are speculative at best and should be minimized.

Table 2 & 3. Sample size per cave varied from 1 to 8. Only those with 1 individual were excluded from Fst calculations. I would suggest setting that minimum at 3 or 4, which, while small, will eliminate a lot of non-significant Fst values from your Tables. Doing so will make Tables 3 smaller and will not change your Results much.

Validity of the findings

28-29 “We identify the climatic changes associated (with) the Messinian Salinity Crisis … as the driving force isolating …” is too strong a statement. Your 95% CIs for those divergence dates span 10 MY (per Figure 4), so drawing a conclusion about a particular event is unreasonable. (And those dates are not even based on confident rate calibrations for these spiders.) Perhaps note they may be associated but anything beyond that suggests a greater degree of confidence in terms of dating of events than is warranted.

This issue needs to be addressed throughout the paper (e.g. line 99 and in the Discussion).

505-535. Revise to emphasize the broad 95% CIs of the early divergence times. This was from 13.5 to 3.5 MYA for the first divergence in Troglohyphantes. That’s a lot of the Miocene!

548-560. Revise conservatively recognizing that Tajima’s D values can be influenced by population structure. And these spiders certainly have impressive population structure.

Additional comments

This paper does a nice job comparing the population genetics of two (or three) cave spiders with overlapping ranges. Note I am not familiar with ENM approaches and so have not provided any comments on those parts of the manuscript.

---

## Round 0.2 · accepted · Accept

· Academic Editor

Accept

You did a very good job replying to the reviewers and the ms is much improved.